# Validity and reliability International Classification of Diseases-10 codes for all forms of injury: A systematic review

**Sarah Paleczny** [iD]*, **Nosakhare Osagie, Jai Sethi, Michael D. Cusimano**

Injury Prevention Research Office, Division of Neurosurgery, St. Michael's Hospital, Toronto, Ontario, Canada

* sarah.paleczny@mail.utoronto.ca

**Data Availability Statement:** All relevant data are within the manuscript and its Supporting Information files.

## Abstract

### Background

Intentional and unintentional injuries are a leading cause of death and disability globally. International Classification of Diseases (ICD), Tenth Revision (ICD-10) codes are used to classify injuries in administrative health data and are widely used for health care planning and delivery, research, and policy. However, a systematic review of their overall validity and reliability has not yet been done.

### Objective

To conduct a systematic review of the validity and reliability of external cause injury ICD-10 codes.

### Methods

MEDLINE, EMBASE, COCHRANE, and SCOPUS were searched (inception to April 2023) for validity and/or reliability studies of ICD-10 external cause injury codes in all countries for all ages. We examined all available data for external cause injuries and injuries related to specific body regions. Validity was defined by sensitivity, specificity, positive predictive value (PPV), and negative predictive value (NPV). Reliability was defined by inter-rater reliability (IRR), measured by Krippendorff's alpha, Cohen's Kappa, and/or Fleiss' kappa.

### Results

Twenty-seven published studies from 2006 to 2023 were included. Across all injuries, the mean outcome values and ranges were sensitivity: 61.6% (35.5%-96.0%), specificity: 91.6% (85.8%-100%), PPV: 74.9% (58.6%-96.5%), NPV: 80.2% (44.6%-94.4%), Cohen's kappa: 0.672 (0.480–0.928), Krippendorff's alpha: 0.453, and Fleiss' kappa: 0.630. Poisoning and hand and wrist injuries had higher mean sensitivity (84.4% and 96.0%, respectively), while self-harm and spinal cord injuries were lower (35.5% and 36.4%, respectively). Transport and pedestrian injuries and hand and wrist injuries had high PPVs (96.5% and 92.0%, respectively). Specificity and NPV were generally high, except for abuse (NPV 44.6%).

**Funding:** This systematic review is supported by the Canadian Institutes of Health Research (CIHR) and the CIHR Canada Graduate Scholarships – Master's (CGS – M) funding (URL: https://www.nserc-crsng.gc.ca/students-etudiants/pg-cs/cgsm-bescm_eng.asp). Also, this review is supported by CIHR (application #471342; URL: https://cihr-irsc.gc.ca/e/193.html). The funding had no role in the study's design, or in the collection, analysis, and interpretation of the data; in the manuscript writing; and in the decision to submit this work for publication. All work was completed independently of the funding agency and all authors take responsibility for the integrity of the work.

**Competing interests:** The authors have declared that no competing interests exist.

## Conclusions and significance

The validity and reliability of ICD-10 external cause injury codes vary based on the injury types coded and the outcomes examined, and overall, they only perform moderately well. Future work, potentially utilizing artificial intelligence, may improve the validity and reliability of ICD codes used to document injuries.

## Background

Injuries are a prevalent issue worldwide, as world-wide deaths due to all injuries has increased from 4,260,493 (uncertainty interval: 4,085,700 to 4,396,138) in 1990 to 4,484,722 (4,332,010 to 4,585,554) in 2017 [1]. Furthermore, all-injury incidence (i.e., new cases) increased from 354,064,302 (338,174,876 to 371,610,802) in 1990 to 520,710,288 (493,430,247 to 547,988,635) in 2017 [1]. Thus, accurate reporting of injuries is critical so healthcare providers, government officials, and policy makers can be informed about injury rates and which types are most prevalent, and for accurate reporting. This allows for an understanding of where public health actions or other healthcare actions may be beneficial to make decisions and take action to prevent injuries and treat them better. Since International Classification of Diseases, Tenth Revision (ICD-10) codes are one of the primary sources of information for reporting diagnoses and are commonly used in research, the analysis of their accuracy is especially important.

ICD codes are used worldwide in all areas of healthcare as a coding system to report diagnoses. In addition to being a coding diagnostic reporting system, they may be used for billing purposes, claims processing, medical care review, classifying data, and for healthcare statistics reporting [2]. The ICD codes are the most widely used classification system for hospital records, and approximately 70% of global health expenditure is distributed according to their data [3, 4]. Therefore, accurate reporting of these codes is essential for maintaining high-quality healthcare data worldwide.

The 10th revision of ICD codes was developed by the World Health Organization (WHO) and is currently used worldwide [3, 4]. These codes have been in effect since approximately year 2000, though this varies by country. A primary use of the ICD-10 codes is for injury data surveillance and research, for which hospital-managed case records are a main source. The injury ICD-10 codes include codes for the external causes of injury conditions (the circumstances and other characteristics of events that led to injury conditions), and the primary injury outcomes themselves.

Despite their wide use in healthcare, the overall validity and reliability of the ICD-10 codes for external-cause injuries has yet to be examined in a systematic review. Individual studies have reported their validity and reliability for different types of injuries, but an overall analysis of the ICD-10 codes' accuracy to diagnose/identify the correct conditions based on how they are coded has not been reported for these outcomes. Thus, there is a gap in the literature reporting the statistics of whether the ICD-10 codes reported in medical records for external cause injuries accurately describe the patients' diagnoses (i.e., the codes' validity), and whether they are coded consistently (i.e., the codes' reliability).

Studies examining the accuracy of external cause of injury ICD-9 codes (E-codes; within the ICD-9th Revision-Clinical Modification (ICD-9-CM)) found that ICD-9-CM-coded data may be able to use broad external cause code blocks with some confidence, while caution should be exercised for very specific code blocks [5, 6]. Nevertheless, ICD-10 external cause codes are very different from ICD-9-CM codes as ICD-10 codes have more specificity and a different structure across code blocks [6].

Our study aims to investigate the validity and reliability of ICD-10 codes for external-cause injuries to report the overall accuracy of these codes in identifying the correct diagnoses (i.e., validity), and whether reporting is reproducible amongst individuals coding them (i.e., reliability). We conducted a systematic review of studies reporting on the validity and/or reliability of ICD-10 codes for classifying patients with intentional and unintentional external injuries including all ages and all countries.

## Methods

### Literature search

An extensive search was conducted in Ovid MEDLINE, EMBASE, COCHRANE, and SCOPUS, from all dates available (1966–2023, 1947–2023, 1996–2023, 1996–2023, respectively). The searches were conducted on the following dates, from database inception to current date: Ovid MEDLINE (April 16/2023), Cochrane Library (April 18/2023), EMBASE (April 18/2023), and Scopus (April 19/2023). The searches ran in each database are available as supplementary materials (S1–S4 Texts). Two reviewers (SP and NO) independently screened the studies. Any disagreements were discussed with a third reviewer (MC). Article screening was completed using Covidence software. Also, a supplementary search of the literature was conducted via the authors manually searching the publications in the reference lists of all relevant articles.

A protocol for this study was published on the International Platform of Registered Systematic Review and Meta-analysis Protocols (https://doi.org/10.37766/inplasy2023.8.0022, [7]). Our review was completed in accordance with the Preferred Reporting Items for Systematic Reviews and Meta-Analyses (PRISMA) framework, and a completed checklist is provided as supplementary information (S1 Checklist).

### Inclusion criteria

Studies that examined validity and/or reliability for the specified ICD-10 injury codes were included in the analysis. All studies included must have been peer reviewed, primary articles, published in English, examining humans, and have full-text available. All ages and countries were included as the ICD-10 codes we investigated are primarily uniform across countries. In studies where only some of the codes examined were ICD-10 injury codes, the relevant results were extracted if they are reported as separate outcome values in the paper.

**Population.** The population examined included patients that experienced an external injury of all ages from any country. The ICD-10 codes used in the inclusion criteria to classify external injuries are summarized in Table 1. This includes resulting injury codes and external cause of the injury codes. Only cases that examined and recorded these injuries with the specified ICD-10 injury codes were included in the analysis.

The ICD-10 codes we selected for our analysis to categorize and present injury data were based on the reliable standards reported by the Association of Public Health Epidemiologists in Ontario (APHEO) [8] and Parachute's 2022 guidelines for ICD-10 code classifications used to document injury causes [9], The codes included are primarily based on the ICD-10-CA codes, as these are applicable to classify injuries in all countries [10]. These injury codes overlap across all countries, with a few minor discrepancies which are described in the results section. We divided the available results into ICD-10 code categories for external causes of injuries (i.e., self-harm injuries, abuse, transport and pedestrian injuries, and poisoning) and injuries to body regions (i.e., hand and wrist injuries, brain injuries, spinal cord injuries, lower extremities injuries, and multiple (total body) injury types reported).

**Table 1. ICD-10 codes used to classify external cause injuries.**

| Unintentional Injuries | |
|---|---|
| **Injury** | **ICD10 Codes** |
| All Unintentional Injuries | V01-X59, Y85-Y86 |
| Cut/pierce | W25-W29, W45, W46 |
| Burns | X00-X19 |
| Exposure to smoke fire/flames | X00-X09 |
| Hot objects/substances | X10-X19 |
| Near-drowning/Submersion | W65-W74, V90, V92 |
| Bathtub | W65, W66 |
| Swimming pool | W67, W68 |
| Natural water | W69, W70 |
| Watercraft | V90, V92 |
| Falls | W00-W19 |
| Unintentional Poisoning | X40-X49 |
| Suffocation, including choking | W75-W84 |
| Overexertion | X50 |
| Natural/Environment | W42, W43, W53-W64, W92-W99, X20-X39, X51-X57 |
| Struck by or against | W20-W22, W50-W52 |
| Motor Vehicle Collisions (Traffic and Non-traffic) | V02-V04, V09.0, V09.2, V12-V14, V19.0–19.2, V19.4-V19.6, V20-79, V80.3–80.5, V80.9, V81.0–81.1, V82.0–82.1, V82.8, V83-V86, V87 (.0-.8), V88 (.0-.8), V89.0, V89.2 |
| Pedestrian | V01-V09 |
| Motor-vehicle Traffic only | V02-V04 (.1, .9), V09.2 |
| Motor-vehicle Non-traffic | V02-V04 (.0), V09 (.0) |
| Other, non-motor vehicle | V01, V05, V06, V09 (.1, .3, .9) |
| Pedal Cycle (Cycling) | V10-V19 |
| Motor-vehicle Traffic only | V12-V14 (.3-.9), V19 (.4-.6) |
| Motor-vehicle Non-traffic | V12-14 (.0-.2), V19 (.2) |
| Other, non-motor vehicle | V10-11, V15-V18, V19 (.3, .8, .9) |
| Public Transportation | V05, V15, V25, V35, V45, V55, V65, V70-79, V81, V82 |
| Bus occupant | V70-79 |
| All railway train or railway vehicle transport accidents | V05, V15, V25, V35, V45, V55, V65, V75, V81 |
| Street car occupant | V82 |
| Off-road transport accidents: (Both traffic* and non-traffic**) | V86 |
| Snowmobiles | V86.00, V86.10, V86.30, V86.50, V86.51, V86.60, V86.61, V86.90, V86.91 (includes drivers, passengers, and unspecified occupants). |
| Other all-terrain or off-road vehicle | V86.08, V86.18, V86.2, V86.38, V86.4, V86.58, V86.68, V86.7, V86.98 (includes drivers, passengers and unspecified occupants). |
| Intentional Injuries | |
| **Injury** | **ICD10 Codes** |
| All intentional injuries | X60-Y09, Y87.0, Y87.1 |
| Self-harm | X60-X84, Y87.0 |
| Assault | X85-Y09, Y87.1 |
| Sports and Recreation | |
| **Description** | **ICD10 codes** |
| Baseball | W22.05, W51.05 |

*(Continued)*

**Table 1.** (Continued)

| Unintentional Injuries | |
|---|---|
| **Injury** | **ICD10 Codes** |
| Hit by ball | W21.00 |
| Hit by bat | W21.01 |
| Cycling | V10-V19 |
| Fall involving rollerblade/scooter/ skateboard | W02.02, W02.03, W02.08 |
| Football/rugby | W22.03, W51.03 |
| Hockey | W21.02, W21.03, W22.02, W51.02 |
| Ice Skates | W02.00 |
| Playground Equipment | Prior to year 2009: the ICD10 code is W09. |
| | From 2009 and onwards: subcategories were introduced and the ICD10 codes are now W09.00-W09.09 |
| Pool and natural water swimming/diving/drowning | W16, W67-W74 |
| Ski/snowboard | W02.01, W02.04, W22.00, W51.00 |
| Soccer | W22.04, W51.04 |
| Tobogganing | W22.01, W51.01 |
| Recreational* boating | V90-V94, only (0.2–0.8) |
| ATV/Snowmobile | V86 |
| ATV (all-terrain or off-road vehicle) | V86.08, V86.19, V86.2, V86.4, V86.5 V86.6, V86.7, V86.9, V86.38, V86.58, V86.68, V86.98 |
| Snowmobile only | V86.00, V86.10, V86.30, V86.50, V86.51, V86.60, V86.90, V86.91 |
| Other sports related injuries | W02.08, W21.08, W21.09, W22.07, W51.07 |
| **Injuries, poisoning and certain other consequences of external causes related to body regions** | |
| **Injury** | **ICD10 Codes** |
| Injuries to the head | S00-S09 |
| Injuries to the neck | S10-S19 |
| Injuries to the thorax | S20-S29 |
| Injuries to the abdomen, lower back, lumbar spine, and pelvis | S30-S39 |
| Injuries to the shoulder and upper arm | S40-S49 |
| Injuries to the elbow and forearm | S50-S59 |
| Injuries to the wrist and hand | S60-S69 |
| Injuries to the hip and thigh | S70-S79 |
| Injuries to the knee and lower leg | S80-S89 |
| Injuries to the ankle and foot | S90-S99 |
| Injuries involving multiple body regions | T00-T07 |
| Injuries to unspecified parts of trunk, limb, or body region | T08-T14 |
| Effects of foreign body entering through natural orifice | T15-T19 |
| Burns and corrosions | T20-T32 |
| Poisoning by drugs, medicaments, and biological substances | T36-T50 |
| Toxic effects of substances chiefly nonmedicinal as to source | T51-T65 |
| Other and unspecified effects of external causes, including abuse and maltreatment (child and adult), temperature-related injuries, asphyxiation, and other unspecified injuries | T66-T78 |
| Alleged physical child abuse | Z61.6 |

(*Continued*)

**Table 1.**  (Continued)

| Unintentional Injuries | |
| --- | --- |
| **Injury** | **ICD10 Codes** |
| Sequelae of injuries, or poisoning, and of other consequences of external cause | T90-T98 |
| Poisoning by exposure to drugs and/or alcohol and/or other toxic chemicals | Y10-Y19 |
| Injuries of any kind with undetermined intent | Y20-Y34 |

**Intervention.**   The intervention evaluated in this review was the validity and/or reliability reported for the specified ICD-10 injury codes.

**Comparator.**   Studies were included which compared the reported ICD-10 injury codes to chart review and/or physician diagnosis as the gold standard (for validity measures) and/or those that compared ICD-10 injury codes between coders or other healthcare workers (i.e., inter-rater reliability (IRR) for reliability measures).

**Outcomes.**   The outcome measures included in the analysis to assess the validity and reliability of external injury ICD-10 codes were: (1) sensitivity, specificity, positive predictive value (PPV), and negative predictive value (NPV) for validity, and (2) IRR, measured by Krippendorff's alpha, Cohen's Kappa, and/or Fleiss' kappa, for reliability.

## Data extraction

Two reviewers (SP and NO) independently reviewed the full-text articles using Covidence software, and any discrepancies were discussed after independent review. A third reviewer (MC) was consulted for extra discussion if necessary. Zotero software was used for extracting the articles once consensus was reached. The PICO (Population, Intervention, Comparator, Outcomes) inclusion framework was utilized for all screening and full-text review to ensure consistency amongst reviewers via a comprehensive checklist on Excel. This framework is commonly used in systematic reviews in healthcare to ensure high quality literature review and results reporting [11]. Thus, papers were screened for the population being injured patients (defined by the external injuries codes listed in Table 1), the intervention being an analysis of ICD-10 codes, the comparator being physician diagnosis and/or chart review, which was evaluated against the recorded ICD-10 codes, and the outcomes being validity (measured as sensitivity, specificity, PPV, and NPV), and reliability (measured as Krippendorff's alpha, Cohen's Kappa, and/or Fleiss' kappa). Only the relevant articles and statistics that met all inclusion criteria were extracted from all papers screened to calculate/report the final summary values.

## Quality assessment

All studies included in our analysis were assessed for risk of bias to investigate study quality using an adaption of the Quality Assessment of Diagnostic Accuracy Studies (QUADAS) tool [12]. Factors such as the study design, patient population, and comparison to the chosen gold standard all may impact the results of the studies included in our paper. Thus, we used the QUADAS protocol to analyze each study and report these findings to be considered when reviewing our results. Furthermore, this method for quality assessment has been previously used in diagnostic accuracy analyses of ICD codes [13, 14].

Each reviewer independently answered the 14 QUADAS questions to assess the quality of all the full-text studies included for these areas of bias. Then, each study was classified as

having a high risk of bias, moderate risk of bias, or low risk of bias based on a qualitative assessment. This classification is consistent with previous studies that used QUADAS to examine ICD codes' diagnostic accuracy [13, 14]. The QUADAS framework used to analyze the studies based on previously published analyses is summarized in S5 Text. Our risk of bias assessment did not include one of the 14 questions from the QUADAS tool (and thus was evaluated out of 13 questions) as it was not applicable to this type of quality assessment. This is consistent with the previous ICD diagnostic accuracy studies completed [12–14].

## Statistical analysis

The outcomes values including sensitivity, specificity, PPV, and NPV for validity, Krippendorff's alpha, Cohen's kappa, and Fleiss' kappa for reliability, were extracted from all papers and used to calculate a summary value. The ICD-10 injury codes from the inclusion criteria were separated into 9 main injury-based categories by grouping similar injury outcomes. Ranges and mean values were calculated and reported for each of the outcomes in all the injury categories to provide an overall estimate of the validity and/or reliability of the ICD-10 codes for those injuries. Means were compared amongst injury categories and totaled for overall estimates of validity and reliability.

Our results calculations averaged all individual studies' outcomes, so all studies were weighted equally. This was done to minimize bias in our results to avoid some studies being weighted heavier simply due to the codes being examined multiple times in different ways. However, for the reporting of values when discussing the studies' bias/quality, all values without the averaging of outcomes were reported to analyze the full spectrum of ranges reported without adjustments. Since sample size was not explicitly reported for all studies (e.g., those where injury patients were a portion of the ICD-10 codes reported and only total sample size was provided) this element was not used for weighting in our statistical analysis.

## Results

### Literature search

We identified 910 records through our original searches (from database inception to April 2023) of the MEDLINE, EMBASE, Cochrane Library, and Scopus databases. Of these, 309 were identified as duplicates, which left 601 articles for title and abstract screening. The search selection framework was completed in accordance with the PRISMA framework, which is summarized in Fig 1 [15]. The full-text reports of 27 articles were sought, but three were excluded due to lack of full-text availability (n = 2) or the article being published in French (n = 1). The remaining 24 were assessed for eligibility, of which four were excluded due to not using ICD-10 codes (n = 1), not using chart review/physician diagnosis as a gold standard for evaluating validity (n = 2), or not calculating outcome measures that exclusively correspond to injuries (n = 1), leaving 20 articles. We also identified 479 records from citation searches. From this search, nine articles' full-text reports were assessed for eligibility, with two excluded for not reporting injuries. Thus, a total of 33 articles were assessed for eligibility, of which six were excluded, leaving 27 articles included for this systematic review of external cause of injury codes.

### Study characteristics

**Demographic variables.** Of the 27 articles that were included in the final review, 13 (48%) were from the United States of America (U.S.A.), six (22%) were from Canada, four (15%) were from Australia, two (7%) were from Taiwan, one (4%) was from Iran, and one (4%) was from Norway. Characteristics of all included studies are presented in Table 2. Sample sizes of

the injury patients' codes included varied widely between the studies and the codes, ranging from the tens in some studies to the thousands in others (S1 Table). The records that were reviewed cover a 38-year period (1982 to 2020), with two (7%) articles that analyzed records between 1982 and 2000, 10 (37%) articles that analyzed records between 2001 and 2010, and 20 (74%) articles that analyzed records between 2011 and 2020.

**Gold standard.** Chart review was used as the gold standard in 22 articles, and direct physician diagnosis based on patient evaluation was used in two articles. The three remaining articles did not use a gold standard, as they only evaluated the IRR of their respective ICD-10 codes of focus.

## Quality assessment

The quality of the included studies was evaluated using the QUADAS tool [12]. Of the 27 studies, 22 (81%) were categorized as high quality, and the remaining 5 (19%) as medium quality (Fig 2). A detailed breakdown of the quality assessment for each study is provided in S2 Table.

## Injury categories

Nine main injury categories were used to report the outcomes of interest based on the relevant literature reported within our inclusion criteria. These include external causes of injuries: self-harm injuries, abuse, transport and pedestrian injuries, and poisoning, and, resulting bodily injuries categorized by body parts: hand and wrist injuries, brain injuries, spinal cord injuries, lower extremities injuries, and multiple (total body) injury types reported (i.e., injury/trauma codes reported as groupings of multiple injury types). A detailed breakdown of all the relevant codes that were included and examined from all studies is listed in S3 Table.

## Statistical outcomes and data analysis

All relevant results and the summary calculations for ranges and mean value per injury category for each outcome are summarized in S4 Table.

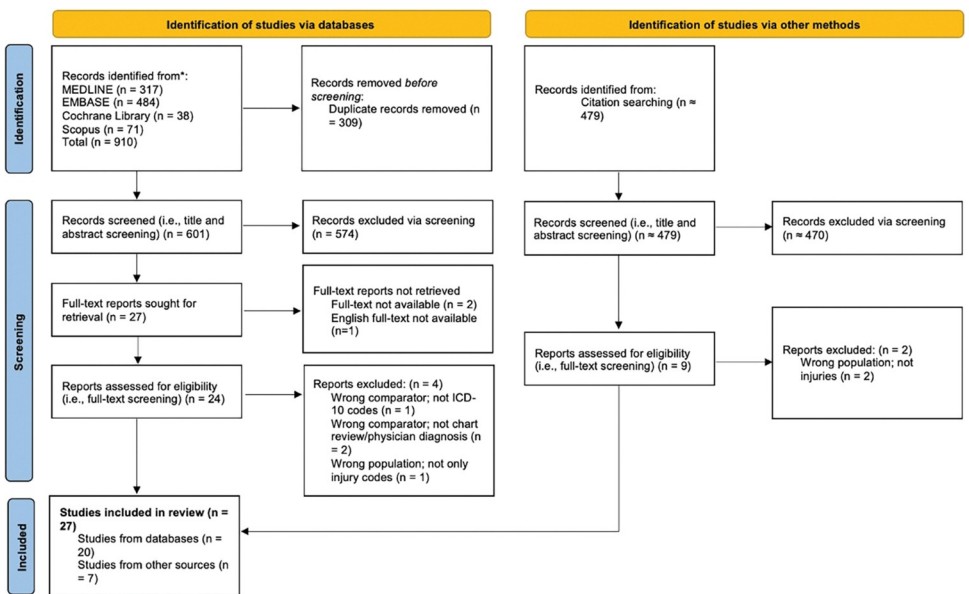

**Fig 1. Diagram of study selection and review.** Preferred Reporting Items for Systematic Reviews and Meta-Analyses (PRISMA)-style Flowchart of Study Selection and Review. Abbreviations: ICD-10 = International Classification of Diseases, Tenth Revision.

**Table 2. Characteristics of studies included.**

| Reference title, author, year | Population | Years of data included | Type of database examined | ICD-10 country |
|---|---|---|---|---|
| **(Karkhaneh et al., 2012)** [16] | All ages, cycling-/pedestrian-related injuries | May to August (2001, 2004, and 2007) | Emergency department (ED) coders assigned codes after reviewing physician-assigned diagnoses at the time of ED discharge (emergency department information system) | ICD-CA |
| **(Sveticic et al., 2020)** [17] | All ages, self-harm | July 1, 2017 to December 31, 2017 | The discharging (from the emergency department) clinician assigned codes (emergency department information system) | ICD-AM |
| **(Furlan & Fehlings, 2011)** [18] | Adults, spinal cord injury | May 2003 to April 2007 | Medical staff assigned codes for diagnoses and clinical interventions (National Trauma Registry) | ICD-CA |
| **(Rasooly et al., 2023)** [19] | Children, physical abuse | Oct 1, 2015 to Sept 30, 2020 | Medical staff assigned codes during inpatient, emergency department, urgent care, and outpatient encounters | ICD-CM (U.S.A.) |
| **(McChesney-Corbeil et al., 2017)** [20] | Children, traumatic brain injury | October 4, 2005 to June 6, 2007 | Trained health technologists coded the discharge abstract database and the inpatient ambulatory care classification system | ICD-CA |
| **(Chiang et al., 2022)** [21] | All ages, carbon monoxide poisoning | 2011 to 2020 | Medical staff assigned coding in the discharge diagnosis (electronic medical records and claims data reported to the National Health Insurance Administration) | ICD-CM (Taiwan) |
| **(Seltzer et al., 2022)** [22] | Adults, ankle fracture | January 1, 2016 to January 1, 2020 | Medical staff assigned coding in the electronic medical records | ICD-CM (U.S. A.) |
| **(Warwick et al., 2020)** [23] | All ages, traumatic brain injury | October 1, 2015 to March 31, 2019 | Medical coders assigned codes using notes and diagnostic statements of the clinical provider (hospital discharge records) | ICD-CM (U.S.A.) |
| **(Schneble et al., 2020)** [24] | Adults, femur fracture | October 1, 2014 to October 1, 2016 | Medical staff assigned codes in the electronic medical records and billing documentation | ICD-CM (U.S.A.) |
| **(Peng et al., 2018)** [25] | All ages, emergency department | October 2013 to December 2013 | Hospital coders coded the charts from emergency visits (hospital discharge abstract data) | ICD-CA |
| **(Watzlaf et al., 2007)** [26] | All ages, public health diagnoses | June 30, 2003 to August 5, 2003 | Medical staff assigned codes for patients from various medical settings (medical records) | ICD-CM (U.S.A.) |
| **(Cheng et al., 2021)** [27] | All ages, adverse drug effect | July 1, 2016 to June 30, 2018 | Medical staff assigned codes for patients upon discharge (hospital claims data) | ICD-CM (Taiwan) |
| **(Thuy Trinh et al., 2018)** [28] | All ages, hip fracture | January 2014 to June 2016 | Medical staff assigned codes upon patient admission (health information exchange) | ICD-AM |
| **(Welk et al., 2014)** [29] | Adults, traumatic spinal cord injury | April 1, 2002 to January 31, 2012 | Medical staff coded diagnoses and clinical interventions (medical records) | ICD-CA |
| **(Peterson et al., 2021)** [30] | All ages, unspecified head injury | October 2015 to December 2018 | Medical staff assigned codes for initial medical encounters that had been discharged (medical records) | ICD-CM (U.S.A.) |
| **(Hagen et al., 2009)** [31] | All ages, traumatic spinal cord injury | 1982 to 2001 | Codes were assigned by the attending physician, and written at the end of the hospitalization (hospital discharge records) | ICD-10 (Norway) |
| **(Randall et al., 2017)** [32] | Adults, self-harm | January 1, 2009 to December 31, 2012 | Medical staff coded individuals admitted to inpatient units after emergency department presentation (hospital discharge abstract database) | ICD-CA |
| **(Hughes Garza et al., 2021)** [33] | Children, physical abuse | 2016–2017 | Codes were assigned after hospital discharge (hospital child abuse registry of multidisciplinary child protection team evaluations) | ICD-CM (U.S.A.) |
| **(Gabella et al., 2022)** [34] | Children, self-harm | Jan 2018 to Dec 2019; Jan 2019 to Dec 2019; Oct 2017 to Sep 2018 | Medical staff assigned coding in the billing records (medical records) | ICD-CM (U.S.A.) |
| **(Miller et al., 2022)** [35] | All ages, firearm injury | October 1, 2015, to December 31, 2019 | Codes were assigned by medical records coders and trauma registrars (in discharge data) | ICD-CM (U.S.A.) |
| **(Brown et al., 2023)** [36] | Children, abusive head trauma | January 1, 2016 to December 31, 2018 | Medical staff assigned codes in the medical records | ICD-CM (U.S.A.) |
| **(McKenzie et al., 2011)** [4] | Children, maltreatment (neglect, physical abuse, sexual abuse, psychological abuse, and other/unspecified abuse) | 2003 to 2006 | Medical staff assigned codes for every hospital discharge (Queensland Health Admitted Patient Data Collection) | ICD-10-AM |

*(Continued)*

**Table 2.** (Continued)

| Reference title, author, year | Population | Years of data included | Type of database examined | ICD-10 country |
|---|---|---|---|---|
| (Green et al., 2017) [37] | All ages, overdose/ poisoning event | 2003 to 2013 | Medical staff assigned codes in death data records | ICD-CM (U.S.A.) |
| (Asadi et al., 2022) [38] | All ages, trauma | 2018 | Medical staff assigned codes in the medical records | ICD-10 (Iran) |
| (Hansen et al., 2021) [39] | Children, self-harm | January 1, 2016 to September 30, 2019 | Medical staff coded emergency department discharge billing records (emergency Department discharge administrative records) | ICD-CM (U.S.A.) |
| (Henderson et al., 2006) [40] | All ages, various injury types | 1998 to 1999, 2000 to 2001 | Medical staff assigned codes for both diagnoses and procedures (public hospital data) | ICD-AM |
| (Shehab et al., 2019) [41] | All ages, adverse drug effect | October 1, 2015 to September 30, 2016 | Medical staff assigned codes for acute care events (hospital administrative claims) | ICD-CM (U.S.A.) |

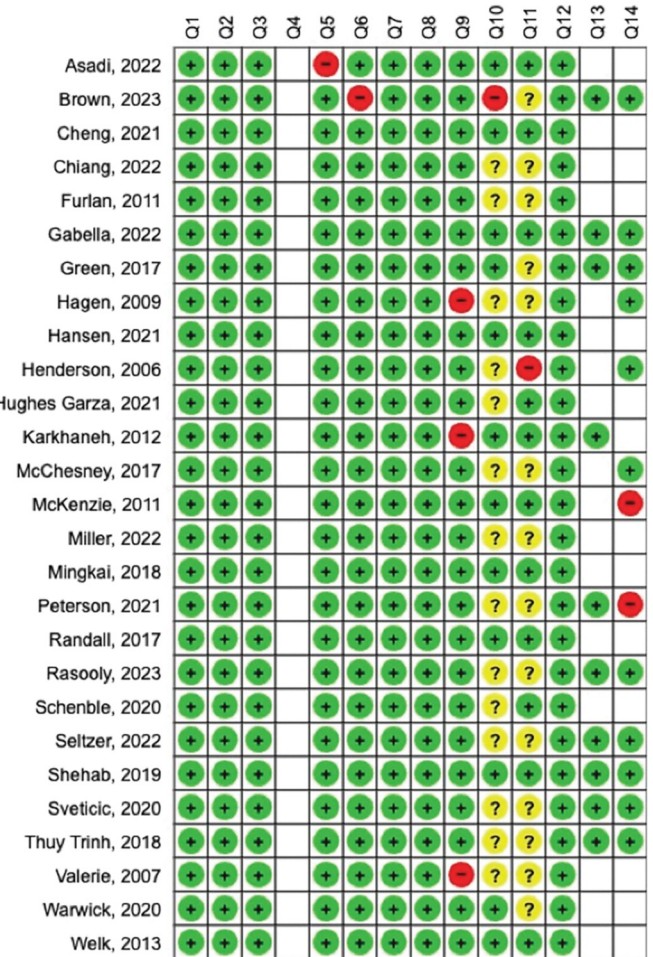

**Fig 2. Quality assessment summary of studies.** The quality of the studies included using the Quality Assessment of Diagnostic Accuracy Studies (QUADAS) tool.

**Sensitivity and specificity.**   Sixteen studies examined sensitivity, with 46 outcome values reported, while 12 studies examined specificity, with 33 outcome values reported for the ICD-10 codes being examined. Across the 9 injury categories, the mean sensitivity was 61.6% (range 35.5%-96.0%), while the mean specificity was 91.6% (range of 85.8%-100%). These values are summarized in Fig 3.

**PPV and NPV.**   In the context of this study, positive predictive values assess the ratio of true positive cases to the total number of cases identified by the ICD-10 codes. Negative predictive values assess the ratio of true negative cases to the total number of cases identified by the ICD-10 codes as not having the condition. Twenty-three studies examined positive predictive values, with 61 outcome values reported, while 9 studies examined negative predictive values, with 20 outcome values reported for the ICD-10 codes of interest. Across the 9 injury categories, the mean positive predictive value was 74.9%, (range 58.6%-96.5%), while the mean negative predictive value was 80.2%, (range of 44.6%-94.4%). The values for each injury category are summarized in Fig 4.

**Inter-rater reliability.**   The inter-rater reliability (IRR) evaluation was conducted using 3 measurement tools: Krippendorff's alpha, Cohen's kappa, and Fleiss' kappa. Nine studies examined Cohen's kappa, resulting in 16 reported outcome values. One study also examined Krippendorff's alpha, with 1 outcome value reported. Another study examined reliability using Fleiss' kappa, reporting 1 outcome. Across the 9 injury categories, the mean Cohen's kappa value was 0.672, (range of 0.480–0.928). With limited data for Krippendorff's alpha and Fleiss' kappa, the values yielded were 0.453 and 0.630, respectively. Fig 5 summarizes the mean and range values for the IRR outcomes in each injury category.

**Injury category statistical analysis.**   The mean and range of each injury outcome are reported in S4 Table.

**Highest-quality study outcomes.**   The results for the 22 studies considered high-quality from our risk of bias assessment are summarized in Table 3. The overall results were: sensitivity 64.5% (35.5%-96.0%), specificity 88.9% (85.83%-100%), PPV: 71.09% (54.92%-92.0%), NPV: 77.82% (44.6%-92.80%), and IRR values: Krippendorff's alpha: 0.453, Cohen's kappa: 0.660 (0.335–0.920), and Fleiss' kappa (0.630). The calculations for these are listed in S4 Table.

## External causes of injuries outcomes

**Transport and pedestrian injuries.**   Two studies assessed transport and pedestrian injuries, resulting in 8 relevant outcome values reported (S1 Table). These studies covered bicycle injuries, pedestrian injuries, femur fractures, and transport incident injuries. Sensitivity was examined by both articles and ranged from 33.1%-95.7% (mean 73.8%). Specificity and positive predictive values were examined by one article each, and resulted in values of 100% and 96.5%, respectively. Cohen's kappa ranged from 0.905–0.945 (mean 0.928). One of the two studies that reported injury mechanism codes for transport and pedestrian injuries was rated as high quality, while the other was rated as medium quality. There was little difference in the inter-rater reliability scores between the studies: inter-rater reliability ranged from 0.91 to 0.98 in the medium-quality study, and from 0.88 to 0.97 in the high-quality study. However, there was an unclear difference in the sensitivity values, which ranged from 87% to 98% in the medium-quality study, but ranged from 25.0% to 45.0% for one half of the values, and from 90.2% to 98.3% for the other half, in the high-quality study.

**Self-harm injuries (Intentional).**   Four studies examined self-harm injuries, with 22 relevant outcome values reported (S1 Table). These 4 studies investigated poisoning, intentional self-harm, and events of undetermined intent. Intentional self-harm included suicide attempts and self-harm of various types, such as poisoning, asphyxiation, and others. Some of these

**INJURY MECHANISMS AND OUTCOMES BY SENSITIVITY VALUES**

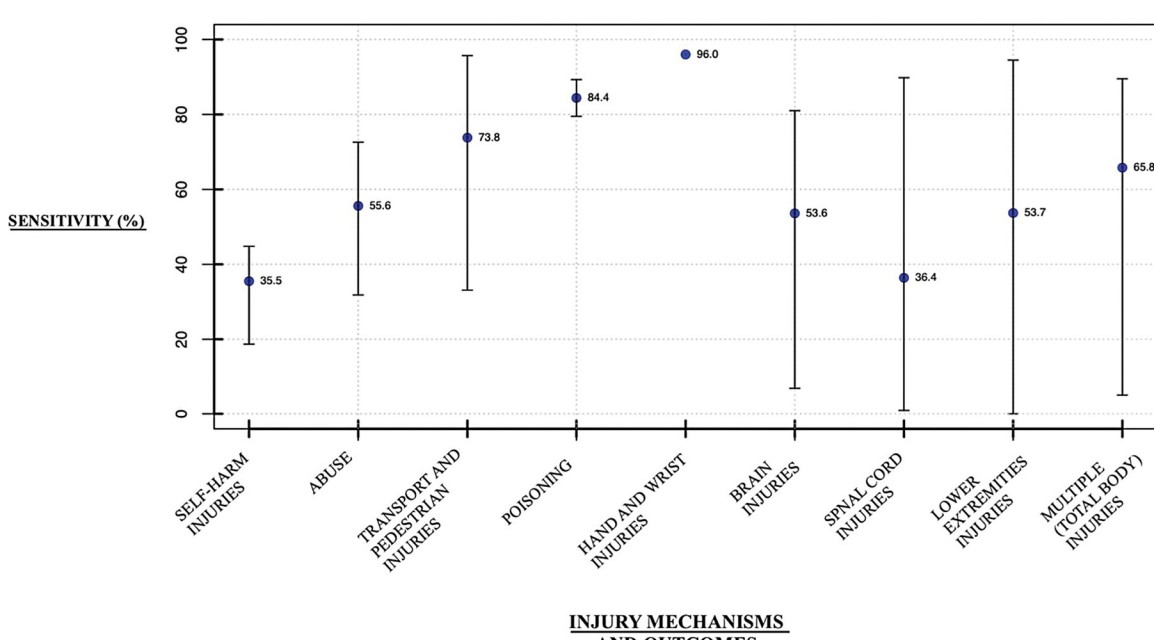

**INJURY MECHANISMS AND OUTCOMES BY SPECIFICITY VALUES**

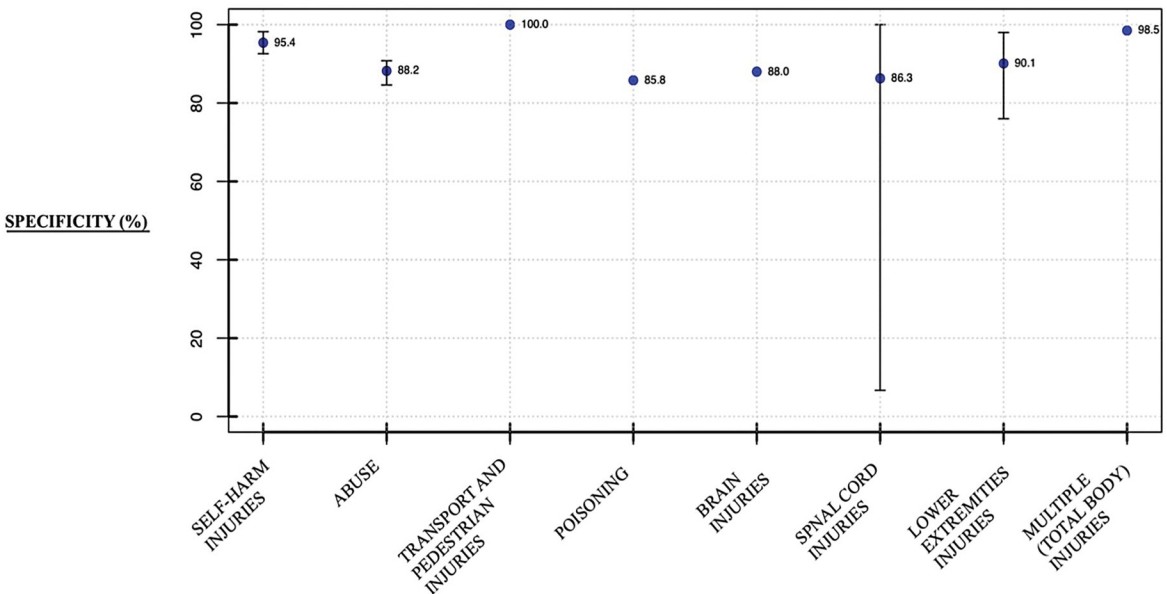

**Fig 3. Sensitivity and specificity outcomes of external cause injury ICD-10 codes.** The mean sensitivities (Panel A) and specificities (Panel B), with error bars reflecting the range of values (where reported), from studies that validated ICD-10 codes for injury mechanisms and outcomes in hospitalization data. **a.** Sensitivity outcomes for all injury categories. **b.** Specificity outcomes for all injury categories.

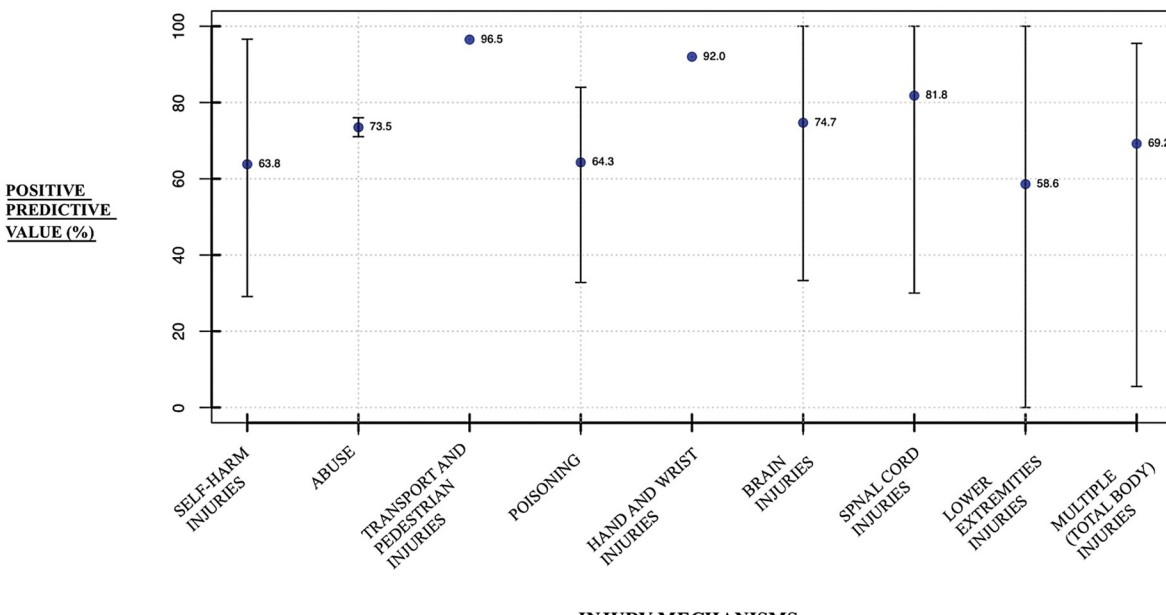

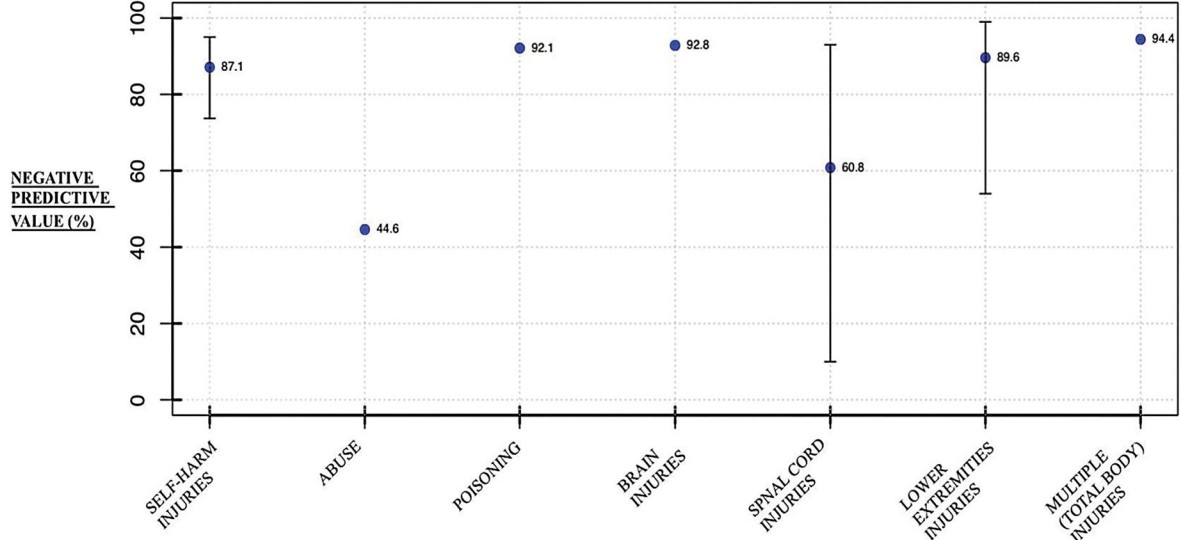

**Fig 4. PPVs and NPVs of external cause injury ICD-10 codes.** The mean PPVs (Panel A) and NPVs (Panel B), with error bars reflecting the range of values (where reported), from studies that validated ICD-10 codes for injury mechanisms and outcomes in hospitalization data. **a.** PPV outcomes for all injury categories. **b.** NPV outcomes for all injury categories.

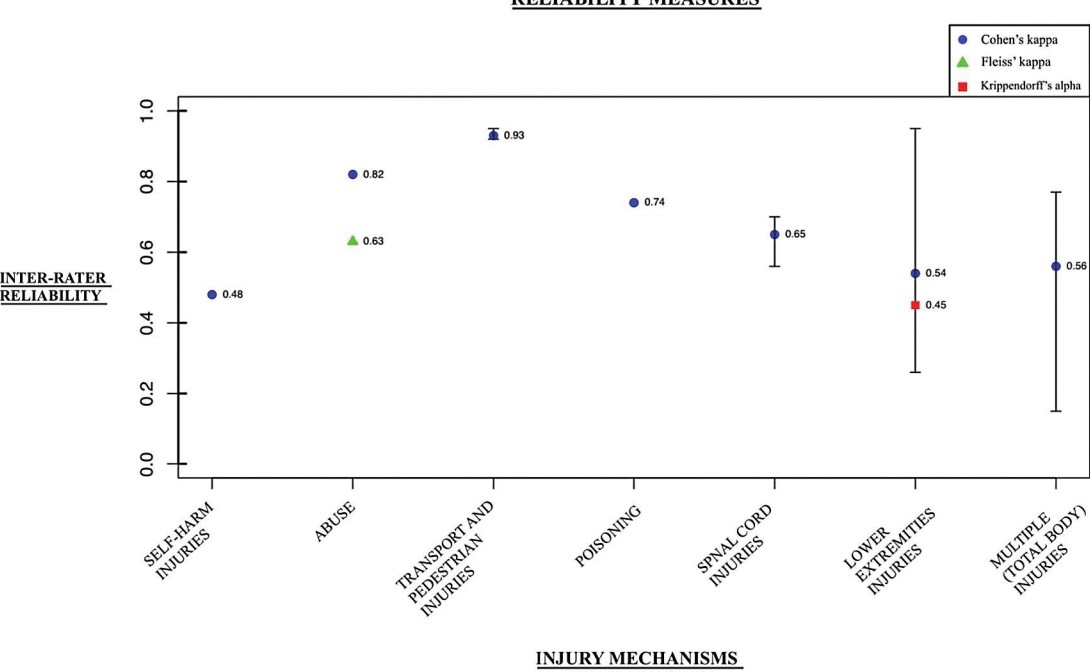

**Fig 5. Inter-rater reliability outcomes for external cause injury ICD-10 codes.** The mean inter-rater reliabilities, with error bars reflecting the range of values (where reported), from studies that analyzed the reliability of ICD-10 codes for injury mechanisms and outcomes in hospitalization data.

studies examined multiple outcomes of interest, while others only reported a few. Sensitivity and specificity ranged from 18.7%-44.8% (mean 35.5%) and 92.6%-98.2% (mean 95.4%), respectively. Positive predictive values ranged from 29.1%-96.6% (mean 63.8%), and negative predictive values ranged from 73.3%-95.0% (mean 87.1%). Cohen's kappa ranged from 0.478 to 0.481 (mean 0.48). All the studies that analyzed self-harm injuries were rated as high-quality.

**Abuse.** The overarching injury topic of abuse explored a range of topics, including child physical abuse, assault, sexual abuse, and other forms of maltreatment. Three studies examined injuries as a result of abuse, with 11 relevant outcome values reported (S1 Table). With each study having a different focus area, it led to a complete and well-rounded exploration of this injury type. Sensitivity and specificity ranged from 31.8%-72.6% (mean 55.6%) and 84.6%-90.8% (mean 88.2%), respectively. Positive predictive values ranged from 71.1–76.0% (mean 73.5%). The assessment of negative predictive values was limited to a single study, resulting in a value of only 44.6%. Similarly, Cohen's kappa and Fleiss' kappa resulted in values of 0.818 and 0.63, respectively. All the studies that reported injury codes for abuse were rated as high-quality.

**Poisoning.** As poisoning is a broad topic, it is important to note that various aspects are included, including toxic effects of carbon monoxide and poisoning from drugs and biological substances. Furthermore, poisoning by opioids, other synthetic narcotics, and psychodysleptics (both intentional and unintentional) were included. Four studies examined injuries caused by poisonings, with 14 relevant outcome values reported (S1 Table). Sensitivity and positive predictive values ranged from 79.5%-89.3% (mean 84.41%) and 32.8%-84.0% (mean 64.3%), respectively. Specificity and negative predictive values were only assessed by one article and

**Table 3. Summary of the validity and reliability outcomes for all high-quality studies included in the analysis.** All values are reported as mean percentage (with ranges) where data was available.

| | | Injury Type | | | | | | | | | Mean (Range) Values Across Injury Types |
| | | Self-harm injuries | Abuse | Transport and Pedestrian Injuries | Poisoning | Hand and wrist injuries | Brain injuries | Spinal cord injuries | Lower extremities injuries | Multiple (total body) injury types reported | |
|---|---|---|---|---|---|---|---|---|---|---|---|
| **Validity Outcomes: % (range)** | Sensitivity | 35.50 (18.7–44.8) | 55.63 (31.8–72.6) | 64.42 (33.12–95.72) | 89.33 | 96.0 | 53.57 (6.8–81) | 72.76 (50–89.8) | 49.62 (0–94) | 63.74 (5.00–89.50) | **64.51 (35.5–96.0)** |
| | Specificity | 95.35 (92.6–98.2) | 88.16 (84.6–90.8) | 100.00 | 85.83 (79.0–98.8) | - | 88.00 | 64.94 (6.7–98) | 90.57 (76–98) | 98.50 | **88.92 (85.83–100)** |
| | PPV | 63.84 (29.10–96.64) | 73.53 (71.06–76) | - | 63.73 (32.8–84) | 92.0 | 77.79 (33.30–100.00) | 78.46 (30.00–90.00) | 54.92 (0–100.00) | 64.48 (5.5–95.5) | **71.09 (54.92–92.0)** |
| | NPV | 87.08 (73.70–95.00) | 44.60 | - | 92.08 (87.9–95.5) | - | 92.80 | 60.80 (10.00–93.00) | 89.57 (54.00–99.00) | - | **77.82 (44.6–92.80)** |
| **Mean (Range) Values Across Validity Outcomes** | | **70.44 (35.50–95.35)** | **65.48 (44.60–88.16)** | **41.11 (0–100)** | **82.7 (63.73–92.08)** | **47.00 (0–96.0)** | **78.04 (53.57–92.8)** | **69.24 (60.80–78.46)** | **71.17 (49.62–90.57)** | **56.68 (0–98.50)** | |
| **Reliability Outcomes** | Krippendorff's alpha | - | - | - | - | - | - | - | 0.453 (0.313–0.593) | - | **0.453** |
| | Cohen's kappa | 0.480 (0.478–0.481) | 0.818 | 0.920 (0.905–0.935) | - | - | - | 0.647 (0.560–0.700) | 0.335 (0.26–0.41) | 0.760 (0.75–0.77) | **0.660 (0.335–0.920)** |
| | Fleiss' kappa | - | 0.630 | - | - | - | - | - | - | - | **0.630** |
| **Mean (Range) Values Across Reliability Outcomes** | | **0.480** | **0.724 (0.630–0.818)** | **0.920** | - | - | - | **0.647** | **0.394 (0.335–0.453)** | **0.760** | |

gave values of 85.8% and 92.1%, respectively. Cohen's kappa yielded a value of 0.735. Three of the four studies that reported injury mechanism codes for poisoning (unintentional) were rated as high quality, while the other was rated as medium quality. There was a difference in the sensitivity values, which ranged from 76% to 83% in the medium-quality study, and from 81.2% to 94.9% among the high-quality studies. However, there was an unclear difference in the PPVs, which ranged from 67% to 71% in the medium quality study but ranged from 32.8% to 60.3% for about half of the values, and from 76.6% to 97.9% for the other half, among the high-quality studies (with PPV ≥ 76.6% in two of the three high-quality studies reporting on PPV).

## Injuries classified by body parts outcomes

**Neurological/spinal cord injuries.** This category covers fractures and nerve injuries, as well as spinal cord injuries. Additionally, it examines outcomes related to injuries of the brain and spinal cord, encompassing concussion, edema, and nerve injuries. Three studies assessed spinal cord injuries, resulting in 47 relevant outcome values reported (S1 Table). Sensitivity and specificity ranged from 0.9%-89.8% (mean 36.4%) and 6.7%-100% (mean 86.3%). Positive predictive values and negative predictive values yielded a range of 30.0%-100% (mean 81.8%) and 10.0%-93.0% (mean 60.8%). Cohen's kappa ranged from 0.56–0.70 (mean 0.65). Two of the three studies that reported injury outcome codes for spinal cord injuries (unintentional) were rated as high quality, while the other was rated as medium quality. There was little

difference in the PPVs, which ranged from 76.2% to 100.0% in the medium-quality study (with the exception of an outlier: 33.3%), and from 76.0% to 97.0% (with the exception of an outlier: 30.0%) among the high-quality studies. There was a difference in the sensitivity values, which ranged from 0.9% to 33.3% in the medium-quality study, and from 50.0% to 89.8% (with the exception of an outlier: 30.0%) among the high-quality studies. However, there was an unclear difference in the specificity values, which ranged from 98.8% to 100.0% in the medium-quality study, but from 6.7% to 25.8% for about half of the values, and from 97% to 98% for the other half, among the high-quality studies.

**Hand and wrist injuries.** Hand and wrist injuries encompasses open wounds on the wrists and hands, along with fractures, sprains, and strains of joints and ligaments. One study examined hand and wrist injuries, with 2 relevant outcome values reported (S1 Table). The values for sensitivity and positive predictive values were 96% and 92%, respectively. This study was rated as high-quality.

**Brain injuries.** This injury category discusses a range of brain injury outcomes, including skull fractures, concussions, cerebral edema, traumatic brain injuries, hemorrhage, and other intracranial injuries. Additionally, it examines outcomes linked to shaken infant syndrome and unspecified head injuries. Five studies assessed brain injuries, resulting in 13 relevant outcome values reported (S1 Table). Sensitivity and positive predictive values ranged from 6.8%-81% (mean 53.6%) and 33.3%-100% (mean 74.7%), respectively. Specificity and negative predictive values were only assessed by one article and gave values of 88.0% and 92.8%, respectively. Four of the five studies that reported injury outcome codes for brain injuries (unintentional) were rated as high quality, while the other was rated as medium quality. There was not a clear difference in the PPVs, which ranged from 22.7% to 73.7% (with about half of the values $\leq$ 40.8%, and the other half $\geq$ 60.3%) in the medium-quality study, and from 60.6% to 100.0% (with the exception of an outlier: 33.3%) among the high-quality studies.

**Lower extremities injuries.** As "lower extremities injuries" is a broad topic, the specific factors that are included are ankle fractures, hip fractures (including proximal femur fractures), as well as fractures, sprains and strains of joints and ligaments at the ankle and foot level. Five studies examined injuries in the lower extremities, resulting in 40 relevant outcome values reported (S1 Table). Sensitivity and specificity ranged from 0%-94.5% (mean 53.7%) and 76.0%-98.2% (mean 90.6%), respectively. Positive predictive values and negative predictive values ranged from 0%-100% (mean 58.6%) and 54.0%-99.0% (mean 89.6%), respectively. Cohen's kappa ranged from 0.26–0.95 (mean 0.54). Krippendorff's alpha was reported in one study and yielded a value of 0.453. Four of the five studies that reported injury outcome codes for lower extremity injuries (unintentional) were rated as high quality, while the other was rated as medium quality. There was a difference in the PPVs, which ranged from 91.0% to 100.0% in the medium-quality study, and from 0.0% to 100.0% (most $\geq$ 43.0%) among the high-quality studies. There was a difference in the sensitivity values, which ranged from 94.0% to 95.0% in the medium-quality study, and from 0.0% to 96.0% (most $\geq$ 50.0%) among the high-quality studies. Inter-reliability scores ranged from 0.93 to 0.97 in the medium-quality study, and from 0.26 to 0.60 among the high-quality studies.

**Multiple (total body) injury types reported.** This injury category is broader but includes critical injury types that are essential for a complete analysis from studies that examined multiple injury-types in one analysis. This category discusses external-cause injuries to different body parts, burns, firearm injuries (both accidental and intentional), head and neck injuries, and trauma codes with unspecified details. Four studies examined multiple (total body) injury types, resulting in 17 relevant outcome values reported (S1 Table). Sensitivity and positive predictive values ranged from 5.0%-89.5% (mean 65.8%) and 5.5%-95.5% (mean 69.2%), respectively. Specificity and negative predictive values were only assessed by one article and yielded

values of 98.5% and 94.4%, respectively. Cohen's kappa ranged from 0.15–0.77 (mean 0.557). Two of the four studies that reported injury outcome codes for multiple (total body) injury types (unintentional) were rated as high quality, while the other two were rated as medium quality. There was little difference in the PPVs, which was 92.6% in the medium-quality study, and ranged from 93.3% to 95.5% (with the exception of an outlier: 5.5%) among the high-quality studies. There was little difference in the sensitivity values, which was 76.1% in the medium-quality study, and ranged from 66.3% to 89.5% (with the exception of an outlier: 5.0%) among the high-quality studies. There was, however, a difference in the inter-reliability scores, which was 0.15 in the medium-quality study, and ranged from 0.75 to 0.77 among the high-quality studies.

## Discussion

### Outcome measures

The values reported across outcomes varied largely depending out the outcome and the injury category, making it difficult to comment on an overall statistic for ICD-10 injury codes, but some key trends in the data stand out. Our findings provide overall summaries for all types of external injuries reported in the literature, as our systematic review is, the first investigation thus far on ICD-10 external injury codes' overall validity and reliability.

**Sensitivity and specificity.** Mean sensitivity values were generally lower (mean = 61.6%, range = 35.5%-96.0%) while specificity values were high across the studies (mean = 91.6%, range = 85.8%-100%). Importantly, due to the nature of sensitivity and specificity measures, when one of these values increases for a diagnostic accuracy test, naturally the other tends to decrease. Thus, a generally high value for both is better but achieving a very high score on both tests is unlikely. Nevertheless, considering the wide use of ICD-10 codes for injury research, these sensitivity values reported are concerning. Furthermore, no gold-standard "cut-off" values have been widely implemented for what are considered to be high- or low-quality values for sensitivity and specificity of diagnostic accuracy studies. In other research contexts, such as Influenza testing, >90% has been reported as excellent sensitivity/specificity, 80–89% is good, 60–79% is fair, while <60% is considered poor [42].

**PPV and NPV.** The overall positive predictive value across the studies was better than the sensitivity values, with a mean of 74.9% (range 58.6%-96.5%), though it was still not very high. The negative predictive value was quite high (mean = 80.0%, range = 44.6%-94.4%). Similarly to sensitivity and specificity, no gold standard "cut-off" has been established for the PPV and NPV of diagnostic accuracy studies. However, in other contexts (e.g., pediatric screening tools), values have been reported for PPV and NPV as: >90%: excellent, 80–89%: good, 60–79%: fair, and <60%: poor [43].

**Inter-rater reliability calculations.** Reliability calculations yielded moderate values, as the mean Cohen's kappa value was 0.672 (range = 0.480–0.928). The limited data for Krippendorff's alpha and Fleiss' kappa were agreement values of 0.453 and 0.63, respectively. Previous reports on these reliability statistics have suggested that 0.81–1.00 is excellent agreement/reliability, 0.61–0.80 is substantial agreement/reliability, 0.40–0.60 is moderate agreement, 0.21–0.39 is fair agreement, and 0.00–0.20 is low agreement (or none) [44, 45].

The wide range of values reported within sensitivity, specificity, PPV, NPV, and inter-rater reliability may be attributable to the large scope of injuries included in the study, as the training of medical personnel, as well as the common coding practices and definitions of codes used to represent injury types within different areas of medicine (e.g., brain injuries compared to self-harm injuries) varies widely. Causes of discrepancies between the outcomes within injury categories are unclear, though differences in sample sizes and study design elements

may be contributing factors (e.g., one researcher versus multiple reviewing the charts). Furthermore, some studies included had a higher risk of bias, as described, which may have impacted the results. Nevertheless, the studies included in our systematic review did have a uniform gold standard of chart review/physician diagnosis and met strict inclusion criteria for the codes and outcomes included.

**Injury outcomes.** Transport and pedestrian injuries, and hand and wrist injuries, had particularly high PPVs (mean = 96.5% and 92.0%, respectively). The rest of the PPVs were moderate to good. The sensitivity values for these categories (transport/pedestrian and hand/wrist injuries) were also quite good (mean = 73.8% and 96.0%, respectively). A common source of misclassification that may have contributed to the good, rather than excellent, sensitivity value, is a lack of training on reporting and classifying pedestrian/transport injuries in neurologist training programs [5, 46] Transport and pedestrian injuries also had particularly high IRR outcomes (mean = 0.93) and specificity values were high for all categories as previously stated. Poisoning codes also had high sensitivity values (mean = 84.4%). The coding discrepancies that may have caused some reduction in this value include that patients with other acute diseases (e.g., burns, and other substance poisoning) may be mistaken for different types of poisoning [21].

Though most sensitivity values were moderate to low, the values for self-harm injuries and spinal cord injuries were particularly low (mean = 36.4% and 35.5%). Some factors that may have contributed to these lower values for self-harm injuries include that a commonly used self-harm code, X84 (intentional self-harm by unspecified means), is more likely to be used in cases of individuals who are Indigenous, those with suicide attempts by cutting, and non-suicidal self-injury in females [32]. Similar bias trends in the reporting of intentional self-harm in different groups, including a bias in reporting of young females self-harm cases in hospital data, have been observed in other studies [47]. The particularly low spinal cord injury sensitivity values may be attributable to the injury characteristics, such as the severity and the level of the spine trauma not being accurately reported in the coding [18]. This was a similar problem found previously in ICD-9 studies that examined the validity and reliability of spinal injuries [48, 49].

The NPVs for abuse were also quite low (mean = 44.6%) compared to the other injury categories, which were all high or relatively high values. This may be attributable to only 5% of the study population receiving the ICD-10-CM code Z04.72 (examination and observation following alleged physical abuse) [33]. The ICD-10-CM guidelines state that all patients should receive this code, though this is not always the case [50]. Furthermore, abuse presents diagnostic challenges as it be inaccurate as proper history is not always taken, and thus may have key omissions [36]. Additionally, abuse may be more difficult for physicians to identify than other conditions, such as traffic injuries, as healthcare professional training in abuse is lacking, especially in certain populations such as elder abuse [51]. Also, there may be reluctances from physicians to diagnose abuse to due to uncertainty and discomfort with these diagnoses [52].

The rest of the outcomes were generally moderate values, including IRR values across the injury categories which were moderate. Of note, there were no results reported for: sensitivity of hand and wrist injuries, NPVs of hand/wrist injuries and transport and pedestrian injuries, and IRR for hand/wrist injuries and brain injuries.

**Causes and solutions for external cause injury coding misclassification.** A variety of sources may have contributed to the lower outcome values reported across the injury categories and the discrepancies amongst coders for the external cause injury ICD-10 codes included in our study as a whole.

Health professionals work under time-constraints, which can lead to errors of omission and commission due to inadequate information for coding or unclear documentation [53].

Incomplete medical histories also may contribute to these errors, so training staff how to best report codes in these cases, as well as emphasizing proper history documentation practices, would be beneficial [36].

Furthermore, inadequate training of hospital staff for coding, and a lack of a standardized approach for ICD-10 coding, which are affected further by variations in staff experience may all contribute to these errors [5]. Thus, more emphasis on training programs that teach accurate coding practices for hospital staff, including admissions staff, providers, and hospital coders could substantially improve common coding misclassifications [5, 46].

## Comparisons of injury ICD-10 coding to non-injury ICD-10 coding

Other non-external injury conditions, such as those for tic disorders and obsessive-compulsive disorder, have been reported to have a PPV of 97% [54]. This high validity may be due to the implementation of increased diagnostic precision of these conditions' psychiatry diagnoses in the Diagnostic and Statistical Manual of Mental Disorders (DSM) resulting in improved diagnostic accuracy [54]. As has been shown from Ruck and colleagues [54], clearer and more detailed specifications for actual ICD-10 diagnoses for external cause injuries may reduce coding errors and improve the validity of these codes. This is especially important for conditions that may have a myriad of physical presentations, such different types of abuse, to improve the accuracy of these codes.

**Future directions.** Although improved physician training programs may have some positive impacts on ICD coding practices, as demonstrated by Paydar & Asadi [55], this alone is likely not enough to significantly improve their validity and reliability. The use of artificial intelligence (AI) for medical record review, such as through natural language processing and deep neural networks to analyze patient files and patient-provided information has been shown to be useful for diagnosis coding practices [56]. Machine learning algorithms can be used to gather chart data and generate codes for diagnoses [57]. For example, Dewaswala and colleagues [58] reported that natural language processing effectively identified and classified hypertrophic cardiomyopathy patients from narrative text reports in cardiac magnetic resonance imaging with high performance compared to manual annotation. Since natural language processing review of medical records can review documents about each patient to reach a diagnostic category, with more development, they may be more efficient and accurate than traditional methods of coding [58].

Thus, implementing these AI algorithms for assigning ICD-10 codes would be beneficial for more accurate coding and to reduce healthcare staff time and energy spent on this. Digital electronic medical records that force clinicians into certain diagnostic categories also hold hope for improving diagnostic and coding accuracy. Some accurate deep learning models have already been created for automatic ICD-10 coding that show promise for the future development of this technology [57]. However, more work is required to integrate these technologies into hospital systems, to train healthcare staff in using them, and to assess the precision of the algorithms' coding before using them regularly. As well, lower income countries, where the major burden of global injury exists, may not have the capacity to introduce expensive electronic medical records.

Furthermore, an analysis investigating the validity and reliability of ICD-10 codes for individual body regions is important to address in future studies, as these codes are also widely used in research and healthcare contexts.

## Quality assessment

### Risks to bias

The risk of differential quality amongst the studies also may have contributed to the discrepancies in the validity and reliability results. In four of the medium-quality studies [30, 31, 36, 40],

it was either unclear or confirmed that the researchers' interpretations of the patient medical charts were not independent of their knowledge of the previously assigned ICD-10 codes, and vice-versa (i.e., whether blinding protocols were utilized). The other medium-quality study [26] did not evaluate the validity of the codes and, therefore, did not use chart review in their analysis. Furthermore, one of the medium-quality studies was vague in the description of the execution of chart review to permit its replication [31], and another did not explain withdrawals from the study [30]. Finally, another study had an additional child abuse scale only used for a portion of the total sample analyzed [36].

**Consistencies in quality.** Despite quality discrepancies, the overall studies' quality were good. All 27 studies met the criteria for good quality regarding seven of the 14 questions from the QUADAS tool (Fig 2). The spectra of patients included in all studies were representative of the patients who would receive the test in practice (i.e., injury patients for each particular injury), reducing the overall risk of spectrum bias [12]. The selection criteria of the studies included were clearly described, and the chart review/physician diagnosis were likely to correctly classify injury patients. Furthermore, the whole samples, or a random selection of patients, received verification (i.e., were compared to chart review/physician diagnosis), reducing the overall risk of partial verification bias. The comparison of chart review/physician diagnosis to ICD-10 codes was done independently (blinded) in some studies, however this was unclear in some.

## Key study strengths

Strengths of our systematic review include the inclusion of all countries in our inclusion criteria for improved generalizability and well-defined, all-encompassing selection criteria for examining ICD-10 external cause injuries. Our study provides key insights for stakeholders who use ICD-10 codes regularly for research, claims processing, health system administration and planning and for policy.

## Limitations

Though our study provides insights into the validity and reliability of ICD-10 external cause injury codes, some limitations exist. There was a large variation in the sample size across the studies, which may have introduced some bias in the results calculations. Furthermore, the inclusion of only English studies may have led to a selection bias and one study did not have an English full-text available. Also, two studies were done prior to 2010, so limited years of ICD-10 data were available. Additionally, a common occurrence with systematic review findings, the amount of data varied for different injury outcomes, depending how much literature was reported for each section. Furthermore, some injury outcomes were only reported in one study (e.g., sensitivity for poisoning and NPV for abuse) which are outlined as injury outcomes without error bars depicting the range in Figs 3–5. This limited the number of results included in the study for some outcomes, which could have introduced bias in the results.

## Conclusion

Injuries are a significant and growing cause of death and disability and have global economic impact. Our results of the validity and reliability of ICD-10 injury codes indicate that caution needs to be exercised in making conclusions when these codes are used for research or policy. While codes such as transport and pedestrian injuries and poisoning had good validity and reliability, others such as those coding for abuse and self-harm require improvement. Strategies such as much more standardized diagnostic criteria for ICD-10 codes and more comprehensive coding training, are required to improve ICD-10 injury coding accuracy. More

widespread use of digital electronic medical records with standardized diagnostic criteria and the use of artificial intelligence techniques that use processes such as natural language processing hold promise for the improvement of coding accuracy and precision into the future.

## Supporting information

**S1 Checklist. Completed PRISMA checklist for systematic reviews.**
(DOC)

**S1 Text. Ovid MEDLINE search (April 16/2023).**
(DOCX)

**S2 Text. Cochrane Library search (April 18/2023).**
(DOCX)

**S3 Text. EMBASE search (April 18/2023).**
(DOCX)

**S4 Text. Scopus search (April 19/2023).**
(DOCX)

**S5 Text. QUADAS framework questionnaire used by all reviewers to assess the studies' quality.**
(DOCX)

**S1 Table. Outcomes of all studies examined.**
(DOCX)

**S2 Table. Risk of bias assessment summary.**
(DOCX)

**S3 Table. Detailed breakdown of all codes examined for each study.**
(DOCX)

**S4 Table. Detailed mean and range calculations for all injury categories and for all validity and reliability outcomes.**
(DOCX)

**S5 Table. Detailed mean and range calculations including only high-quality studies for all injury categories and for all validity and reliability outcomes.**
(DOCX)

## Author Contributions

**Conceptualization:** Sarah Paleczny, Nosakhare Osagie, Jai Sethi, Michael D. Cusimano.

**Data curation:** Sarah Paleczny, Nosakhare Osagie, Jai Sethi, Michael D. Cusimano.

**Formal analysis:** Sarah Paleczny, Jai Sethi, Michael D. Cusimano.

**Funding acquisition:** Sarah Paleczny, Michael D. Cusimano.

**Investigation:** Sarah Paleczny, Nosakhare Osagie, Jai Sethi, Michael D. Cusimano.

**Methodology:** Sarah Paleczny, Nosakhare Osagie, Jai Sethi, Michael D. Cusimano.

**Project administration:** Sarah Paleczny, Michael D. Cusimano.

**Resources:** Sarah Paleczny, Nosakhare Osagie, Jai Sethi, Michael D. Cusimano.

**Software:** Sarah Paleczny, Jai Sethi, Michael D. Cusimano.

**Supervision:** Sarah Paleczny, Michael D. Cusimano.

**Validation:** Sarah Paleczny, Nosakhare Osagie, Michael D. Cusimano.

**Visualization:** Sarah Paleczny, Nosakhare Osagie, Michael D. Cusimano.

**Writing – original draft:** Sarah Paleczny, Nosakhare Osagie, Jai Sethi, Michael D. Cusimano.

**Writing – review & editing:** Sarah Paleczny, Nosakhare Osagie, Jai Sethi, Michael D. Cusimano.

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
