## [Decision Letter · Decision Letter 0]

21 Dec 2023

PONE-D-23-34822Validity and reliability International Classification of Diseases-10 codes for all forms of injury: a systematic reviewPLOS ONE

Dear Dr. Paleczny,

Thank you for submitting your manuscript to PLOS ONE. After careful consideration, we feel that it has merit but does not fully meet PLOS ONE’s publication criteria as it currently stands. Therefore, we invite you to submit a revised version of the manuscript that addresses the points raised during the review process.

We look forward to receiving your revised manuscript.

Kind regards,

Hans-Peter Simmen, M.D., Professor of Surgery

Academic Editor

PLOS ONE

Journal Requirements:

Additional Editor Comments:

It was a difficult task to identify and find reviewers. I am happy to have two experts making the review. Your paper is important and well written. However, the reviewers recommend a minor revision. Please, address their comments..

Reviewers' comments:

Reviewer's Responses to Questions

**Comments to the Author**

1. Is the manuscript technically sound, and do the data support the conclusions?

Reviewer #1: Yes

Reviewer #2: Yes

2. Has the statistical analysis been performed appropriately and rigorously? 

Reviewer #1: Yes

Reviewer #2: Yes

3. Have the authors made all data underlying the findings in their manuscript fully available?

Reviewer #1: Yes

Reviewer #2: Yes

4. Is the manuscript presented in an intelligible fashion and written in standard English?

Reviewer #1: Yes

Reviewer #2: Yes

5. Review Comments to the Author

Reviewer #1: PLOS ONE- RE: Validity and reliability International Classification of Diseases-10 codes for all forms of injury: a systematic review

Thank you for the opportunity to review this paper:

The aim of this systematic review was to investigate the validity and reliability of ICD-10 codes for external-cause injuries.

This is important to gain a broader knowledge of the overall accuracy of these codes in identifying the correct diagnoses and also of the reproducibility among different coding individuals. The study showed that the validity and reliability of ICD-10 external cause injury codes vary based on the injury types coded and the outcomes examined, and overall, they only performed moderately well. It also highlights the need for future studies in order to increase the performance of the ICD codes (AI, etc.).

I congratulate the authors for the present important work, particularly because it’s the first overall analysis (systematic review) of the validity and reliability of ICD-10 codes for external-cause injuries.

The methodology, presentation and interpretation of the data presented is adequate and relevant. I only have a few comments to clarify:

Introduction:

Nothing to add.

Methods:

- Personally, I would recommend to remove the definitions of sensitivity, specificity, PPV and NPV (lines 170-180). It makes it hard to read and is somehow obvious.

- - In/exclusion criteria: Consider listing the criteria for either exclusion or inclusion, but not in both sections, as this is redundant for the reader (peer-reviewed, publications in English, full text available,etc.)

Results:

- Line 243-249: Consider to rephrase this section. For the reader it’s confusing that you mention that a total of 33 articles were assessed for eligibility, whereby six were excluded. After the explanation why you excluded those 6 articles you mention another three articles that were excluded from the original search. However, finally you state that 27 articles were included for this systematic review of external cause of injury codes (33-6-3=24).

Discussion:

- Good structure of discussion, nothing to add.

Figures and Tables

- Provide them in sufficient quality. The diagrams provided for review were blurry.

Reviewer #2: Thank you for the opportunity to review this manuscript submitted to PLOS ONE.

As stated by the authors, ICD codes provide the basis for a number of clinical applications. Furthermore, ICD codes may serve as a data source for research purposes. The authors thus addressed an important topic in patients suffering from injuries.

The manuscript is well written and clearly structured. The methods are adequate to investigate the validity and reliability of ICD-10 codes for external cause injuries

I have a few minor comments:

1. Page 4, line 71: The abbreviation “UI” should be written out in full.

2. Page 6, line 35: “All studies included must have been peer reviewed, primary articles, published in English, examining humans, and have full-text available.” As stated by the authors, ICD-10 codes are used worldwide. The inclusion of artciles in English only may have led to a selectio bias. This should be mentioned in the limitations.

3. Page 11, line 11: In my understanding, PICOS would rather be structured as follows: Population: injured patients, intervention: ICD-10 coding, comparator: physician diagnosis/chart review, outcome: validity (measured as sensitivity, specificity, PPV, and NPV) and reliablity (Krippendorff's alpha, Cohen’s Kappa, and/or Fleiss’ kappa). Please explain the structure of the PICOS framework in more detail.

4. Page 13, line 222: “Ranges and mean values were calculated and reported for each of the outcomes in all the injury categories to provide an overall estimate of the validity and/or reliability of the ICD-10 codes for those injuries.” Usually the standard deviation is given for the mean and the range or interquartile range for the median. What was the rationale to report the mean with a range?

5. Results, Table 3: As it seems, some outcomes were only reported in one study. (e.g., sensitivity for poisoning, specificity for brain injuries, PPV for hand and wrist injuries, and NPV for abuse). I would suggest mentioning this as an additional limitation in the discussion section.

6. The discussion section starts with the key study strengths. As a reader, I would be more interested in the key findings of the study at the beginning of the discussion. The authors may want to consider discussing the main findings of this review at the beginning of the discussion.

7. This review investigated the validity and reliablity of ICD-10 codes for external cause injuries. It would also be interesting to investigate the validity and reliability of ICD-10 codes for specific injuries by body region. I understand that this is outside the scope of this review. However, the topic could be addressed in future studies.

6. PLOS authors have the option to publish the peer review history of their article (what does this mean?). If published, this will include your full peer review and any attached files.

Reviewer #1: No

Reviewer #2: No

---

## [Author Response · Author response to Decision Letter 0]

19 Jan 2024

Dear Dr. Chenette and Dr. Simmen,

We would like to thank you for allowing us the chance to amend and re-submit our article for potential publication in your reputable journal PLOS One. We have undertaken the suggested changes and introduced several revisions. We amended the limitations section, the future directions, and clarified the PICOS framework and the inclusion/exclusion criteria. We generated the figures with clearer quality and attached these documents. Additionally, we addressed the questions asked by the reviewers.

Please find below the point-by-point reply to the reviewers.

Please let me know if you have any other comments or suggestions.

Sincerely,

Michael D. Cusimano MD, PhD

Professor of Neurosurgery Education and Public Health

St. Michael’s Hospital, University of Toronto

Toronto, ON, Canada

Reviewer Comments:

Reviewer #1: PLOS ONE- RE: Validity and reliability International Classification of Diseases-10 codes for all forms of injury: a systematic review

Methods:

- Personally, I would recommend to remove the definitions of sensitivity, specificity, PPV and NPV (lines 170-180). It makes it hard to read and is somehow obvious.

We thank the reviewer for their comments and for the review of our study. We have removed these definitions.

- - In/exclusion criteria: Consider listing the criteria for either exclusion or inclusion, but not in both sections, as this is redundant for the reader (peer-reviewed, publications in English, full text available,etc.)

We have removed the exclusion criteria description and kept only the inclusion criteria paragraph.

Results:

- Line 243-249: Consider to rephrase this section. For the reader it’s confusing that you mention that a total of 33 articles were assessed for eligibility, whereby six were excluded. After the explanation why you excluded those 6 articles you mention another three articles that were excluded from the original search. However, finally you state that 27 articles were included for this systematic review of external cause of injury codes (33-6-3=24).

We have amended this section for clarity, which is outlined on page 13, lines 258-266.

Figures and Tables

- Provide them in sufficient quality. The diagrams provided for review were blurry.

We have provided the diagrams with improved quality for better readability. 

We thank the reviewer for their review of our study.

Reviewer #2: Thank you for the opportunity to review this manuscript submitted to PLOS ONE.

1. Page 4, line 71: The abbreviation “UI” should be written out in full.

We thank the reviewer for their comments and for the review of our study. We have amended this accordingly. 

2. Page 6, line 35: “All studies included must have been peer reviewed, primary articles, published in English, examining humans, and have full-text available.” As stated by the authors, ICD-10 codes are used worldwide. The inclusion of artciles in English only may have led to a selectio bias. This should be mentioned in the limitations.

We have added this to the limitations section which can be found on page 33, line 675-676.

3. Page 11, line 11: In my understanding, PICOS would rather be structured as follows: Population: injured patients, intervention: ICD-10 coding, comparator: physician diagnosis/chart review, outcome: validity (measured as sensitivity, specificity, PPV, and NPV) and reliablity (Krippendorff's alpha, Cohen’s Kappa, and/or Fleiss’ kappa). Please explain the structure of the PICOS framework in more detail.

We have re-structured the PICOS framework as suggested and explained this in more detail, so this section is clearer (page 11, lines 184-188).

4. Page 13, line 222: “Ranges and mean values were calculated and reported for each of the outcomes in all the injury categories to provide an overall estimate of the validity and/or reliability of the ICD-10 codes for those injuries.” Usually the standard deviation is given for the mean and the range or interquartile range for the median. What was the rationale to report the mean with a range?

We reported the mean with a range rather than the standard deviation since as mentioned, multiple injury outcomes only had one study which examined that specific injury outcome. Thus, standard deviation was not applicable to these outcomes. For the other injury outcomes, in our opinion, with a small number of outcomes included in the calculations (all <13 outcomes for each injury category), standard deviation would not add to these outcomes. However, the range was reported to provide more information on the range of outcomes included in the mean calculations to show the scope of values included. This is outlined in the summary figures for each outcome where the ranges were reported with error bars depicting the range of outcomes included in the mean calculations (if multiple outcomes were reported). 

5. Results, Table 3: As it seems, some outcomes were only reported in one study. (e.g., sensitivity for poisoning, specificity for brain injuries, PPV for hand and wrist injuries, and NPV for abuse). I would suggest mentioning this as an additional limitation in the discussion section.

We have amended the limitations accordingly and included this, which can be found on pages 33 and 34, lines 680-686.

6. The discussion section starts with the key study strengths. As a reader, I would be more interested in the key findings of the study at the beginning of the discussion. The authors may want to consider discussing the main findings of this review at the beginning of the discussion.

We have amended the discussion by discussing the main findings at the beginning of this section and moving the key study strengths after the discussion on page 33, lines 666-671.

7. This review investigated the validity and reliablity of ICD-10 codes for external cause injuries. It would also be interesting to investigate the validity and reliability of ICD-10 codes for specific injuries by body region. I understand that this is outside the scope of this review. However, the topic could be addressed in future studies.

We agree that this would be interesting to investigate, and we have added this to the future directions section on page 32, lines 638-640.

We thank the reviewer for their review of our study.

---

## [Editor Report · Decision Letter 1]

25 Jan 2024

Validity and reliability International Classification of Diseases-10 codes for all forms of injury: a systematic review

PONE-D-23-34822R1

Dear Dr. Paleczny,

Congratulations for this excellent revision. We’re pleased to inform you that your manuscript has been judged scientifically suitable for publication and will be formally accepted for publication once it meets all outstanding technical requirements.

Kind regards,

Hans-Peter Simmen, M.D., Professor of Surgery

Academic Editor

PLOS ONE
---

## [Editor Report · Acceptance letter]

17 Feb 2024

PONE-D-23-34822R1 

PLOS ONE

Dear Dr. Paleczny, 

I'm pleased to inform you that your manuscript has been deemed suitable for publication in PLOS ONE. Congratulations! Your manuscript is now being handed over to our production team.

Kind regards, 

on behalf of

Dr. Hans-Peter Simmen 

Academic Editor

PLOS ONE